# Robustness of 3D Navier–Stokes System with Increasing Damping

## Jie Cao and Keqin Su *

College of Information and Management Science, Henan Agricultural University, Zhengzhou 450046, China
* Correspondence: keqinsu@hotmail.com

**Abstract:** The principal objective of the paper is the study of the three-dimensional Navier–Stokes system with non-autonomous perturbation force term and increasing damping term, which often appears in the fluid system within saturated porous media and other complex media. With some suitable assumptions on the system parameters and external force term, based on the known result on global well-posedness, the existence of pullback attractors is educed, and the system robustness is shown via the upper semicontinuity of system attractors as the perturbation parameter approaches a certain value.

**Keywords:** Navier–Stokes equations; increasing damping; robustness





## 1. Introduction

The Navier–Stokes system depicts the conservation law about mass and momentum of fluid, reflects the basic relationship among gravity, pressure and other factors inside the viscous fluid, and is widely applied in many fields, such as materials, weather forecast and so on. There has been a good deal of interesting research results on Navier–Stokes equations, such as well-posedness of system solution, asymptotical behaviour, existence and dimension evaluation of several attractors, determination, invariant measures and so on, which can be referred to [1–9].

Given that the 3D NS system is extremely complex, especially in the case of solution well-posedness, the 3D NS system with damping becomes one of the hot research topics. About the increasing damping form $|u|^{\gamma-1}u$, it is necessary for us to study the porous media. Porous media is very common in our lives, such as coal, animal fur, aluminium foams, ceramics and so on. Further, the applications of porous media involve many aspects of our lives—heat transfer device design, sound testing in medicine, food drying, underground water flow, ice melting, sound propagation in building materials, noise reduction in automotive design, heat retention in materials, etc. The porosity is the most representative concept in studying porous media, which is the ratio of fluid volume to the total, a nonnegative number less than or equal to 1, where the fluid is the compound of air and some liquids. From the Darcy law, we know, in the flow within porous media, the flow speed $u$ and the pressure $p$ satisfy

$$\mu u = -k\nabla p,$$

where the viscosity coefficient $\mu$ and permeability coefficient $k$ are both positive. Further, the flow in saturated porous media can be described as

$$\mu u = -k\nabla p + \rho f,$$

where $\rho$ denotes the fluid density in porous media and $f$ is the force. When the flow speed in porous media becomes large to a certain extent, to obtain an accurate description for porous media, the left term $\mu u$ will be replaced by a new one, such as $au + bu^2 + cu^3$.

Therefore, the increasing damping form $|u|^{\gamma-1}u$ appears naturally in the fluid system within the porous media, some detailed conclusions can be referred to [10]. Further, we can find some meaningful results on models with the increasing damping for non-Newtonian fluid in [11–13], and also the simulations on a system with the damping form $|u|^{\gamma-1}u$ is one of the future research directions, where the ideas in [14,15] can be used for reference.

In this paper, we investigate the following three-dimensional incompressible NS system with the increasing damping $|u|^{\gamma-1}u$

$$\begin{cases} u_t - \nu\Delta u + (u \cdot \nabla)u + \beta|u|^{\gamma-1}u + \nabla p = \varepsilon g(x,t), \ (x,t) \in \Omega \times (\varsigma, +\infty), \\ divu = 0, \ (x,t) \in \Omega \times (\varsigma, +\infty), \\ u(x,t)|_{\partial\Omega} = 0, \ t \in (\varsigma, +\infty), \\ u(x,\varsigma) = u(\varsigma), \ x \in \Omega, \end{cases} \tag{1}$$

where $\Omega \subset \mathbb{R}^3$ is a bounded domain with a smooth boundary, $u = u(x,t)$ is the unknown velocity field, $p$ is the pressure, $\nu > 0$ is the viscosity coefficient, $\varepsilon g(x,t)$ is the external force and $\varepsilon > 0$, $\beta > 0$, $\gamma \geq 1$. Meaningful results on system (1) can be found in [16–20]. Where the well-posedness of the strong solution was shown in [16,19], based on which three types of attractors were given in [17,18,20].

In practice, due to measurement errors and circumstances, the parameters of the system are inevitably perturbed. In this case, the study of robustness is particularly important, and robustness has become one of the important indicators of control system design. For the conclusion of studying the robustness through upper semicontinuity, we could refer to [21–25]. Thus far, there is no result about the robustness of the system (1), and the aim of this paper is to study the upper semicontinuity of the pullback attractor to (1).

The general outline is demonstrated as follows. The second and third parts recall some basic definitions, theorems, frequently used Sobolev spaces and assumptions and show the main results. Based on the well-posedness and estimation of the solution, we obtain a pullback attractor of (1) and verify the robustness through semicontinuity in the fourth part.

## 2. Preliminaries

### 2.1. Some Definitions and Theorems

Consider a Banach space $\mathcal{X}$ with metric $d_{\mathcal{X}}(\cdot, \cdot)$ and norm $\| \cdot \|_{\mathcal{X}}$. $\{S(t,\varsigma)\}_{t\geq\varsigma}$ is a process in $\mathcal{X}$, its definition and properties can be referred to in [26].

**Definition 1.** *Suppose that $\tilde{\mathcal{A}}(t)$ is a family of compact subsets $\{\tilde{A}(t)\}_{t\in\mathbb{R}}$, and satisfies*
*(i) Characteristic of invariance: $S(t,\varsigma)\tilde{A}(\varsigma) = \tilde{A}(t)$, $\forall t \geq \varsigma$.*
*(ii) Characteristic of pullback attraction: for any subset $\tilde{D}$ bounded in $\mathcal{X}$, there holds*

$$\lim_{\varsigma\to+\infty} dist_{\mathcal{X}}(S(t,t-\varsigma)\tilde{D}, \tilde{A}(t)) = 0,$$

*then we call $\tilde{\mathcal{A}}(t) = \{\tilde{A}(t)\}_{t\in\mathbb{R}}$ the pullback attractor.*

**Definition 2.** *For arbitrary subset $\tilde{D}$ bounded in $\mathcal{X}$, if there always exists $\bar{T}(t,\tilde{D}) > 0$ satisfying*

$$S(t,t-\varsigma)\tilde{D} \subset \tilde{D}(t), \ \varsigma \geq \bar{T}(t,\tilde{D}), \ t \in \mathbb{R},$$

*then we call the subset family $\tilde{\mathcal{D}} = \{\tilde{D}(t)\}_{t\in\mathbb{R}}$ the pullback absorbing set of $\{S(t,t-\varsigma)\}$.*

**Definition 3.** *For arbitrary $t \in \mathbb{R}$, let $\tilde{\mathcal{D}} = \{\tilde{D}(t)\}_{t\in\mathbb{R}}$ be a subset family in $\mathcal{X}$. For arbitrary sequence $\{\varsigma_n\}$ with $\varsigma_n \to +\infty$ $(n \to +\infty)$ and $\xi_n \in \tilde{D}(t-\varsigma_n)$, if $\{S(t,t-\varsigma_n)\xi_n\}$ possesses the relative compactness in space $\mathcal{X}$, then we call $\{S(t,t-\varsigma)\}$ pullback $\tilde{\mathcal{D}}-$asymptotically compact in space $\mathcal{X}$.*

**Theorem 1.** *Suppose that $\tilde{\mathcal{D}} = \{\tilde{D}(t)\}_{t\in\mathbb{R}}$ is the pullback-absorbing set for $\{S(t, t - \varsigma)\}$, which is continuous and possesses the pullback $\tilde{\mathcal{D}}-$asymptotical compactness in $\mathcal{X}$ topology. Then the subset family $\tilde{\mathcal{A}}(t) = \{\tilde{A}(t)\}_{t\in\mathbb{R}}$, where*

$$\tilde{A}(t) = \Lambda(\tilde{\mathcal{D}}, t) = \bigcap_{s\geq 0} \overline{\bigcup_{\varsigma\geq s} S(t, t - \varsigma)\tilde{D}(t - \varsigma)}^{\mathcal{X}},$$

*is the pullback attractor to $\{S(t, t - \varsigma)\}$ in space $\mathcal{X}$.*

**Definition 4.** *Let $\varepsilon \in (\varepsilon_0 - \hbar, \varepsilon_0 + \hbar)$, $\{\tilde{D}_\varepsilon(t)\}_{t\in\mathbb{R}} \subset \mathcal{X}$ denotes a subset family in space $\mathcal{X}$, and function $\Phi(t, \cdot, \cdot, \cdot, \cdot)$ $(t \in \mathbb{R})$ is defined on*

$$(\varepsilon_0 - \hbar, \varepsilon_0 + \hbar) \times (\varepsilon_0 - \hbar, \varepsilon_0 + \hbar) \times \bigcup_{\varepsilon\in(\varepsilon_0-\hbar,\varepsilon_0+\hbar)} \tilde{D}_\varepsilon(t) \times \bigcup_{\varepsilon\in(\varepsilon_0-\hbar,\varepsilon_0+\hbar)} \tilde{D}_\varepsilon(t).$$

*If for any sequences $\{\varepsilon_n\}_{n\in\mathbb{N}} \subset (\varepsilon_0 - \hbar, \varepsilon_0 + \hbar)$ and $\{\xi_n\}_{n\in\mathbb{N}} \subset \tilde{D}_{\varepsilon_n}(t)$, there always exist subsequences $\{\varepsilon_{n_k}\}_{k\in\mathbb{N}}$ and $\{\xi_{n_k}\}_{k\in\mathbb{N}}$ such that*

$$\lim_{k\to\infty}\lim_{l\to\infty} \Phi(t, \varepsilon_{n_k}, \varepsilon_{n_l}, \xi_{n_k}, \xi_{n_l}) = 0,$$

*then the function $\Phi(t, \cdot, \cdot, \cdot, \cdot)$ is said to be contractive.*

**Definition 5.** *Let $dist_\mathcal{X}(\cdot, \cdot)$ be the Hausdorff semidistance in $\mathcal{X}$, if there holds*

$$\text{(H)} \quad \lim_{\varepsilon\to\varepsilon_0} dist_\mathcal{X}(\tilde{A}_\varepsilon(t), \tilde{A}_{\varepsilon_0}(t)) = 0, \ \forall t \in \mathbb{R},$$

*then it is said that $\tilde{\mathcal{A}}_\varepsilon(t)$ is upper semi-continuous at $\varepsilon_0$.*

**Theorem 2** ([27]). *Let $\varepsilon_n, \varepsilon_0 > 0$, $x_n, x \in \mathcal{X}$, $\varepsilon_n \to \varepsilon_0$, $x_n \to x$ $(n \to +\infty)$, and for arbitrary $t \in \mathbb{R}$ and $\varsigma \in \mathbb{R}^+$ there hold*

    (i) $\lim\limits_{n\to\infty} S_{\varepsilon_n}(t, t - \varsigma)x_n = S_{\varepsilon_0}(t, t - \varsigma)x$.

    (ii) $\exists \bar{T} = \bar{T}(t) > 0$ and $\theta \in (0, 1)$ satisfying

$$U_{\varepsilon_0}(t - \theta\varsigma, t - \varsigma)(\overline{\bigcup_{\varepsilon_0-\hbar<\varepsilon\leq\varepsilon_0+\hbar} \tilde{A}_\varepsilon(t - \varsigma)}^{\mathcal{X}}) \subset \tilde{D}_{\varepsilon_0}(t - \theta\varsigma), \ \forall \varsigma > \bar{T}.$$

    (iii) $\bigcup\limits_{\varepsilon\in(\varepsilon_0-\hbar,\varepsilon_0+\hbar)} \tilde{A}_\varepsilon(t)$ *possesses the relative compactness in $\mathcal{X}$.*

    *Then (H) holds.*

**Lemma 1.** *Let $\varepsilon \in (\varepsilon_0 - \hbar, \varepsilon_0 + \hbar)$. For arbitrary $t \in \mathbb{R}$ and $\delta_0 > 0$, if there exist positive constant $\bar{T}$, depending on $t, \delta_0$ and $\{\tilde{A}_\varepsilon(t)\}$, and a contractive function $\Phi(t - \bar{T}, \cdot, \cdot, \cdot, \cdot)$ with domain*

$$(\varepsilon_0 - \hbar, \varepsilon_0 + \hbar) \times (\varepsilon_0 - \hbar, \varepsilon_0 + \hbar)$$
$$\times \bigcup_{\varepsilon\in(\varepsilon_0-\hbar,\varepsilon_0+\hbar)} \tilde{A}_\varepsilon(t - \bar{T}) \times \bigcup_{\varepsilon\in(\varepsilon_0-\hbar,\varepsilon_0+\hbar)} \tilde{A}_\varepsilon(t - \bar{T}),$$

*such that, for any $\varepsilon_1, \varepsilon_2 \in (\varepsilon_0 - \hbar, \varepsilon_0 + \hbar)$, $\xi \in \tilde{A}_{\varepsilon_1}(t - \bar{T})$ and $\eta \in \tilde{A}_{\varepsilon_2}(t - \bar{T})$, there holds*

$$\|S_{\varepsilon_1}(t, t - \bar{T})\xi - S_{\varepsilon_2}(t, t - \bar{T})\eta\|_\mathcal{X} \leq \delta_0 + \Phi(t - \bar{T}, \varepsilon_1, \varepsilon_2, \xi, \eta),$$

*then $\bigcup\limits_{\varepsilon\in(\varepsilon_0-\hbar,\varepsilon_0+\hbar)} \tilde{A}_\varepsilon(t)$ is relatively compact in $\mathcal{X}$.*

*2.2. Some Sobolev Spaces*

Denote

$$\mathcal{E} := \{u^1 | u^1 \in (C_0^\infty(\Omega))^3, div u^1 = 0\}, \ \mathcal{H} = \overline{\mathcal{E}}^{(L^2(\Omega))^3}, \ \mathcal{V} = \overline{\mathcal{E}}^{(H^1(\Omega))^3},$$

where $\mathcal{H}$ and $\mathcal{V}$ are Hilbert spaces with inner product and norm

$$(u^1, u^2) = \sum_{j=1}^{3} \int_\Omega u_j^1(x) u_j^2(x) dx, \ |u^1| = (u^1, u^1)^{1/2}, \ \forall \, u^1, u^2 \in \mathcal{H},$$

and

$$((u^1, u^2)) = \sum_{i,j=1}^{3} \int_\Omega \frac{\partial u_j^1}{\partial x_i} \frac{\partial u_j^2}{\partial x_i} dx, \ \|u^1\| = ((u^1, u^1))^{1/2}, \ \forall \, u^1, u^2 \in \mathcal{V}.$$

$\mathcal{V} \hookrightarrow \mathcal{H} \equiv \mathcal{H}' \hookrightarrow \mathcal{V}'$, where $\mathcal{V}'$ is the dual space of $\mathcal{V}$ with norm $\|\cdot\|_*$ and dual product $\langle \cdot, \cdot \rangle$. Specifically, $\|\cdot\|_p$ denotes the norm of $(L^p(\Omega))^3$.

$P$ is the orthonormal projection in space $(L^2(\Omega))^3$ to $\mathcal{H}$, and some properties and spectral analysis for the operator $A := -P\Delta$ could be found in [28], where we know $A$ possesses the eigenvalues $\{\lambda_j\}_{j=1}^\infty$ and eigenfunctions $\{\omega_j\}_{j=1}^\infty$, which make an orthonormal system. For any $u^1, u^2 \in \mathcal{V}$, there holds $\langle Au^1, u^2 \rangle = ((u^1, u^2))$. The power $A^s$ is defined as follows

$$A^s f = \sum_j \lambda_j^s a_j \omega_j, \ s \in \mathbb{C}, \ j \in \mathbb{R}, \ f = \sum_j a_j \omega_j,$$

$$D(A^s) = \{f : A^s f \in H\} = \{f = \sum_j a_j \omega_j : \sum_j \lambda_j^{Res} |a_j|^2 < +\infty\},$$

and denote $\overline{\mathcal{E}}^{D(A^s)}$ by $D(A^s)$ with norm $|A^s u|$.

The properties of the bilinear operator $B(u^1, u^2) := P((u^1 \cdot \nabla)u^2)$ and the trilinear operator

$$b(u^1, u^2, u^3) = \langle B(u^1, u^2), u^3 \rangle = \sum_{i,j=1}^{3} \int_\Omega u_i^1 \frac{\partial u_j^2}{\partial x_i} u_j^3 dx,$$

can be found in [6], such as

$$b(u^1, u^2, u^2) = 0, \ b(u^1, u^2, u^3) = -b(u^1, u^3, u^2), \ \forall \, u^1, u^2, u^3 \in \mathcal{V}.$$

In particular, $B(u^1) = B(u^1, u^1)$.

*2.3. Assumptions*

**Definition 6.** *For any $w \in \mathcal{V}$ and $t > \varsigma$, we call $u$ a strong solution to system* (1) *on $\Omega \times [\varsigma, T]$, if $u \in L^\infty(\varsigma, T; \mathcal{V}) \cap L^2(\varsigma, T; D(A)) \cap L^\infty(\varsigma, T; (L^{\gamma+1}(\Omega))^3)$, and satisfies*

$$\begin{cases} \frac{d}{dt}(u, w) + \nu((u, w)) + b(u, u, w) + (P\beta|u|^{\gamma-1}u, w) = (P\varepsilon g, w), \\ u(x, \varsigma) = u(\varsigma). \end{cases} \tag{2}$$

Let $G(u) = P\beta|u|^{\gamma-1}u$, system (1) can be replaced by the following abstract form

$$\begin{cases} u_t + \nu Au + B(u) + G(u) = P\varepsilon g(x, t), \ \forall \, t > \varsigma, \\ u(x, \varsigma) = u(\varsigma). \end{cases} \tag{3}$$

To derive the well-posedness of the solution, let the following conditions hold

(I) $\exists K > 0$ satisfying

$$\sup_{t\in\mathbb{R}} |g(x,t)| < K,\ \sup_{t\in\mathbb{R}} \int_t^{t+1} |g_t|^2 ds < +\infty.$$

(II) $\forall t \in \mathbb{R}$, there holds

$$\int_{-\infty}^t e^{\delta s}|g(x,s)|^2 ds < \infty,\ 0 < \delta < \nu\lambda_1/2.$$

## 3. Main Results

**Theorem 3.** *Let* $3 < \gamma \leq 5$, $u(\varsigma) \in \mathcal{V} \cap (L^{\gamma+1})^3$, *then the strong solution $u$ to system* (1) *exists uniquely, and*

$$\sup_{\varsigma\leq t\leq T} (\|u(t)\|^2 + \|u(t)\|_{\gamma+1}^{\gamma+1}) \leq Ce^{C(T-\varsigma)} \times (\|u(\varsigma)\|^2 + \|u(\varsigma)\|_{\gamma+1}^{\gamma+1} + C\varepsilon^2 \int_\varsigma^T |g|^2 dt).$$

The proof can be referred to in Theorem 3.1 in [20]. Similarly, we can show that the system solution $u$ possesses the continuity to the initial datum and then obtain the existence of a family of processes $\{S_\varepsilon(t,\varsigma) : \mathcal{V} \to \mathcal{V}\}$.

**Theorem 4.** *Let* $3 < \gamma \leq 5$, $u(\varsigma) \in \mathcal{V} \cap (L^{\gamma+1})^3$, *then for any $t \in \mathbb{R}$, system* (3) *possesses the pullback attractor* $\tilde{\mathcal{A}}_\varepsilon(t) = \{\tilde{A}_\varepsilon(t)\}_{t\in\mathbb{R}}$ *in* $\mathcal{V}$.

**Theorem 5.** *Let* $3 < \gamma \leq 5$, $u(\varsigma) \in \mathcal{V} \cap (L^{\gamma+1})^3$, *then for any $t \in \mathbb{R}$, the pullback attractor* $\tilde{\mathcal{A}}_\varepsilon(t)$ *is upper semi-continuous at $\varepsilon_0$, and*

$$\lim_{\varepsilon\to\varepsilon_0} dist_X(\tilde{A}_\varepsilon(t), \tilde{A}_{\varepsilon_0}(t)) = 0.$$

## 4. Proof

### 4.1. Estimation of Solutions

**Lemma 2.** *Let* $3 < \gamma \leq 5$, $u(\varsigma) \in \mathcal{V} \cap (L^{\gamma+1})^3$, *and $t \in \mathbb{R}$ is arbitrary. Then, there exist $T_0 > 0$ and $\rho_1(t) > 0$ satisfying*

$$|u(t)|^2 \leq \rho_1^2(t),\ \varsigma < -T_0.$$

**Proof.** Multiplying (3) by $e^{\delta t}u$, we can get

$$\frac{d}{dt}(e^{\delta t}|u(t)|^2) + K_1 e^{\delta t}(\|u\|^2 + \|u\|_{\gamma+1}^{\gamma+1}) \leq \frac{\varepsilon^2}{\nu\lambda_1}e^{\delta t}|g|^2, \tag{4}$$

where $K_1 = \min\{\nu - \frac{\delta}{\lambda_1}, 2\beta\}$, then integrating it over $[\varsigma, t]$, we also get

$$e^{\delta t}|u(t)|^2 + K_1 \int_\varsigma^t e^{\delta s}(\|u(s)\|^2 + \|u(s)\|_{\gamma+1}^{\gamma+1})ds \leq \frac{\varepsilon^2}{\nu\lambda_1} \int_{-\infty}^t e^{\delta s}|g|^2 ds + e^{\delta\varsigma}|u(\varsigma)|^2. \tag{5}$$

Finally, there holds

$$|u(t)|^2 \leq e^{\delta(\varsigma-t)}|u(\varsigma)|^2 + \frac{\varepsilon^2}{\nu\lambda_1}e^{-\delta t} \int_{-\infty}^t e^{\delta s}|g|^2 ds,$$

which means there is a positive constant $T_0$ satisfying

$$|u(t)|^2 \leq \frac{2\varepsilon^2}{\nu\lambda_1}e^{-\delta t} \int_{-\infty}^t e^{\delta s}|g|^2 ds \equiv \rho_1^2(t),\ \forall\ \varsigma \leq -T_0. \tag{6}$$

$\square$

**Lemma 3.** *Let $3 < \gamma \leq 5$, $u(\varsigma) \in \mathcal{V} \cap (L^{\gamma+1})^3$, and $t \in \mathbb{R}$ is arbitrary. Then, there are positive constants $T_1$ and $\rho_2(t)$ satisfying*

$$\|u(t)\|^2 + \|u(t)\|_{\gamma+1}^{\gamma+1} \leq \rho_2^2(t), \; \varsigma \leq -T_1.$$

**Proof.** Multiplying (3) by $u_t$ and $Au$, respectively, we have

$$2|u_t|^2 + \nu\frac{d}{dt}\|u\|^2 + \frac{2\beta}{\gamma+1}\frac{d}{dt}\|u\|_{\gamma+1}^{\gamma+1}$$
$$\leq 2|((u \cdot \nabla)u, u_t)| + 2|(u_t, \varepsilon g)| \leq |u_t|^2 + 2|u \cdot \nabla u|^2 + 2|\varepsilon g|^2 \qquad (7)$$

and

$$\frac{d}{dt}\|u\|^2 + 2\nu|Au|^2 + 2\beta||u|^{\gamma-1}\nabla u|^2 + \frac{8\beta(\gamma-1)}{(\gamma+1)^2}|\nabla|u|^{\frac{\gamma+1}{2}}|^2$$
$$\leq 2|((u \cdot \nabla)u, Au)| + 2|(Au, \varepsilon g)| \leq \frac{\nu}{2}|Au|^2 + \frac{4}{\nu}|u \cdot \nabla u|^2 + \frac{4}{\nu}|\varepsilon g|^2. \qquad (8)$$

Using the estimates in [20] that

$$C|u \cdot \nabla u|^2 \leq \beta||u|^{\gamma-1}\nabla u|^2 + \frac{\nu}{2}|Au|^2 + C\|u(t)\|_{\gamma+1}^{\gamma+1},$$

and adding (7) and (8) together, we get

$$(\nu+1)\frac{d}{dt}\|u\|^2 + \frac{2\beta}{\gamma+1}\frac{d}{dt}\|u(t)\|_{\gamma+1}^{\gamma+1} + 2\nu|Au|^2 + 2\beta||u|^{\gamma-1}\nabla u|^2 + |u_t|^2$$
$$\leq \beta||u|^{\gamma-1}\nabla u|^2 + \frac{\nu}{2}|Au|^2 + C\|u(t)\|_{\gamma+1}^{\gamma+1} + (\frac{4}{\nu}+2)\varepsilon^2|g|^2,$$

and

$$\frac{d}{dt}(e^{\delta t}(\|u\|^2 + \|u\|_{\gamma+1}^{\gamma+1})) + Ce^{\delta t}(|Au|^2 + ||u|^{\gamma-1}\nabla u|^2)$$
$$\leq Ce^{\delta t}(\|u(t)\|^2 + \|u(t)\|_{\gamma+1}^{\gamma+1}) + C\varepsilon^2 e^{\delta t}|g|^2.$$

By integration of the inequality and conclusion in Lemma 2, we conclude that

$$e^{\delta t}(\|u(t)\|^2 + \|u(t)\|_{\gamma+1}^{\gamma+1})$$
$$\leq e^{\delta\varsigma}(\|u(\varsigma)\|^2 + \|u(\varsigma)\|_{\gamma+1}^{\gamma+1}) + C\int_{\varsigma}^{t} e^{\delta s}(\|u(s)\|^2 + \|u(s)\|_{\gamma+1}^{\gamma+1})ds + C\varepsilon^2\int_{-\infty}^{t} e^{\delta s}|g|^2 ds$$
$$\leq Ce^{\delta\varsigma}(\|u(\varsigma)\|^2 + \|u(\varsigma)\|_{\gamma+1}^{\gamma+1}) + C_1\varepsilon^2\int_{-\infty}^{t} e^{\delta s}|g|^2 ds + C_2,$$

and there are positive constants $T_1$ and $\rho_2(t)$ satisfying

$$\|u(t)\|^2 + \|u(t)\|_{\gamma+1}^{\gamma+1} \leq \rho_2^2(t), \; \forall \varsigma \leq -T_1.$$

$\square$

**Lemma 4.** *Let $3 < \gamma \leq 5$, $u(\varsigma) \in \mathcal{V} \cap (L^{\gamma+1})^3$, and $t \in \mathbb{R}$ is arbitrary. Then, there are positive constants $T_2$ and $\rho_3(t)$ satisfying*

$$|Au(t)|^2 \leq \rho_3^2(t), \; \varsigma \leq -T_2.$$

The proof can be referred to in [19].

### 4.2. Existence of Pullback Attractor

According to the estimations on the system solution, we get

$$\|u(t)\|^2 + \|u(t)\|_{\gamma+1}^{\gamma+1}$$
$$\leq Ce^{-\delta(\varsigma-t)}(\|u(\varsigma)\|^2 + \|u(\varsigma)\|_{\gamma+1}^{\gamma+1}) + C_1\varepsilon^2 e^{-\delta t}\int_{-\infty}^{t} e^{\delta s}|g|^2 ds + C_2. \qquad (9)$$

We denote $R_\varepsilon(t) = (2C_1\varepsilon^2 e^{-\delta t}\int_{-\infty}^{t} e^{\delta s}|g|^2 ds + 2C_2)^{\frac{1}{2}}$, $\tilde{D}_\varepsilon(t) = \{u \in \mathcal{V} : \|u\| \leq R_\varepsilon(t)\}$, then the subset family $\tilde{\mathcal{D}}_\varepsilon(t) = \{\tilde{D}_\varepsilon(t)\}_{t\in\mathbb{R}}$ is just the pullback absorbing set of system (3) in $\mathcal{V}$.

Choose $0 < \delta < \delta_1 < \nu\lambda_1/2$, and replace $\delta$ with $\delta_1$, then (9) still holds. For any $u(t-\varsigma) \in \tilde{D}_\varepsilon(t-\varsigma)$, there holds

$$\|S_\varepsilon(t, t-\varsigma)u(t-\varsigma)\|^2 + \|S_\varepsilon(t, t-\varsigma)u(t-\varsigma)\|_{\gamma+1}^{\gamma+1}$$
$$\leq Ce^{-\delta_1\varsigma}R_\varepsilon^2(t-\varsigma) + C_1\varepsilon^2 e^{-\delta_1 t}\int_{-\infty}^{t} e^{(\delta_1-\delta)s}e^{\delta s}|g|^2 ds + C_2$$
$$\leq Ce^{-\delta_1\varsigma}R_\varepsilon^2(t-\varsigma) + C_1\varepsilon^2 e^{-\delta t}\int_{-\infty}^{t} e^{\delta s}|g|^2 ds + C_2.$$

Since $\lim\limits_{\varsigma\to+\infty} e^{-\delta_1\varsigma}R_\varepsilon^2(t-\varsigma) = 0$, there is $T_3 = T_3(t, \tilde{\mathcal{D}}_\varepsilon(t)) > 0$ satisfying

$$S_\varepsilon(t, t-\varsigma)\tilde{D}_\varepsilon(t-\varsigma) \subset \tilde{D}_\varepsilon(t), \ \forall\, \varsigma > T_3.$$

Lemma 4 and the fact that $D(A) \hookrightarrow\hookrightarrow \mathcal{V}$ lead to that $\{S_\varepsilon(t, \varsigma)\}$ possesses the pullback asymptotical compactness in $\mathcal{V}$. Finally, we obtain Theorem 4 from Theorem 1.

### 4.3. Robustness

Next, the robustness of the system is obtained by showing that the pullback attractors $\tilde{\mathcal{A}}_\varepsilon(t) = \{\tilde{A}_\varepsilon(t)\}_{t\in\mathbb{R}}$ are upper semi-continuous.

**Lemma 5.** *Let $3 < \gamma \leq 5$, $u(\varsigma) \in \mathcal{V} \cap (L^{\gamma+1})^3$, and $\varsigma > 0$. For arbitrary $t \in \mathbb{R}$, $\varepsilon_n \to \varepsilon_0$ and $u_{n,t-\varsigma}^0 \to u_{t-\varsigma}^0$ as $n \to +\infty$, then*

$$\lim_{n\to\infty} S_{\varepsilon_n}(t, t-\varsigma)u_{n,t-\varsigma}^0 = S_{\varepsilon_0}(t, t-\varsigma)u_{t-\varsigma}^0.$$

**Proof.** Let

$$z^{\varepsilon_n}(t) = S_{\varepsilon_n}(t, t-\varsigma)u_{n,t-\varsigma}^0 - S_{\varepsilon_0}(t, t-\varsigma)u_{t-\varsigma}^0 = u^{\varepsilon_n}(t) - u^{\varepsilon_0}(t),$$

then $z^{\varepsilon_n}(t)$ satisfies

$$z_t^{\varepsilon_n} + \nu A z^{\varepsilon_n} + B(u^{\varepsilon_n}) - B(u^{\varepsilon_0}) + G(u^{\varepsilon_n}) - G(u^{\varepsilon_0}) = P(\varepsilon_n - \varepsilon_0)g(x, t), \qquad (10)$$

where $z^{\varepsilon_n}(t-\varsigma) = u_{n,t-\varsigma}^0 - u_{t-\varsigma}^0$.

Multiplying (10) by $z^{\varepsilon_n}(t)$, there is

$$\frac{1}{2}\frac{d}{dt}|z^{\varepsilon_n}|^2 + \nu\|z^{\varepsilon_n}\|^2 + \beta||u^{\varepsilon_n}|^{\frac{\gamma-1}{2}}|z^{\varepsilon_n}||^2$$
$$\leq \int_\Omega |z^{\varepsilon_n}|^2|\nabla u^{\varepsilon_0}|dx + \beta\int_\Omega |z^{\varepsilon_n}||u^{\varepsilon_0}||u^{\varepsilon_n}|^{\gamma-1}$$
$$- |u^{\varepsilon_0}|^{\gamma-1}|dx + (\varepsilon_n - \varepsilon_0)|(g, z^{\varepsilon_n})|. \qquad (11)$$

Using the technique employed in Theorem 3.1 in [20], for $3 < \gamma \leq 5$, we can obtain

$$\frac{d}{dt}|z^{\varepsilon_n}|^2 \leq C(1 + |u^{\varepsilon_n}|^{\frac{\gamma-1}{2}} + |u|^{\frac{\gamma-1}{2}})|z^{\varepsilon_n}|^2 + C(\varepsilon_n - \varepsilon_0)|g|^2. \tag{12}$$

Applying the Gronwall inequality to (12) yields

$$|z^{\varepsilon_n}(t_0)|^2 \leq C(|u^0_{n,t_0-\varsigma} - u^0_{t_0-\varsigma}|^2 + (\varepsilon_n - \varepsilon_0)\int_{t_0-\varsigma}^{t_0}|g|^2 ds)$$

$$\times exp(\int_{t_0-\varsigma}^{t_0}(1 + |u^{\varepsilon_n}|^{\frac{\gamma-1}{2}} + |u^{\varepsilon_0}|^{\frac{\gamma-1}{2}})ds) \to 0 \ (n \to \infty).$$

□

**Lemma 6.** *Let $3 < \gamma \leq 5$, $u(\varsigma) \in \mathcal{V} \cap (L^{\gamma+1})^3$, and $t \in \mathbb{R}$ is arbitrary. Then there are positive constants $\theta \in (0,1)$ and $T_4$, depending on $\varepsilon_0$ and $t$, such that*

$$S_{\varepsilon_0}(t - \theta\varsigma, t - \varsigma)\overline{\underset{\varepsilon_0-\hbar<\varepsilon\leq\varepsilon_0+\hbar}{\cup}\tilde{A}_\varepsilon(t-\varsigma)}^X \subset \tilde{D}_{\varepsilon_0}(t-\theta\varsigma), \ \forall \varsigma \geq T_4.$$

**Proof.** For the pullback attractors $\tilde{\mathcal{A}}_\varepsilon(t) = \{\tilde{A}_\varepsilon(t)\}_{t\in\mathbb{R}}$, we have

$$\tilde{A}_\varepsilon(t) = \underset{s\geq 0}{\cap}\overline{\underset{\varsigma\geq s}{\cup}S_\varepsilon(t, t-\varsigma)\tilde{D}_\varepsilon(t-\varsigma)}^X,$$

and

$$\overline{\underset{\varepsilon_0-\hbar<\varepsilon\leq\varepsilon_0+\hbar}{\cup}\tilde{A}_\varepsilon(t)}^X \subset \overline{\underset{\varepsilon_0-\hbar<\varepsilon\leq\varepsilon_0+\hbar}{\cup}\tilde{D}_\varepsilon(t)}^X \subset \tilde{D}_{\varepsilon_0}(t).$$

Let $\theta = 1 - \delta\delta_1^{-1}$, for any $u^0_{t-\varsigma} \in \tilde{D}_{\varepsilon_0+\hbar}(t-\varsigma)$, applying Lemma 3, we know that

$$\|S_{\varepsilon_0}(t-\theta\varsigma, t-\varsigma)u^0_{t-\varsigma}\|^2 + \|S_{\varepsilon_0}(t-\theta\varsigma, t-\varsigma)u^0_{t-\varsigma}\|^{\gamma+1}_{\gamma+1}$$

$$\leq Ce^{-\theta_1(1-\delta)\varsigma}R^2_{\varepsilon_0+\hbar}(t-\varsigma) + C_1\varepsilon_0^2 e^{-\delta_1(t-\theta\varsigma)}\int_{-\infty}^{t-\theta\varsigma}e^{\delta_1 s}|g|^2 ds + C_2$$

$$\leq Ce^{-\theta_1(1-\delta)\varsigma}R^2_{\varepsilon_0+\hbar}(t-\varsigma) + C_1\varepsilon_0^2 e^{-\delta_0(t-\theta\varsigma)}\int_{-\infty}^{t-\theta\varsigma}e^{\delta_0 s}|g|^2 ds + C_2,$$

and from $\underset{\varsigma\to+\infty}{\lim}e^{-\theta_1(1-\delta)\varsigma}R^2_{\varepsilon_0+\hbar}(t-\varsigma) = 0$ we also get

$$S_{\varepsilon_0}(t-\theta\varsigma, t-\varsigma)\overline{\underset{\varepsilon_0-\hbar<\varepsilon\leq\varepsilon_0+\hbar}{\cup}\tilde{A}_\varepsilon(t-\varsigma)}^X \subset \tilde{D}_{\varepsilon_0}(t-\theta\varsigma).$$

□

**Lemma 7.** *Let $3 < \gamma \leq 5$, $u(\varsigma) \in \mathcal{V} \cap (L^{\gamma+1})^3$, and $t \in \mathbb{R}$ is arbitrary. Then, $\underset{\varepsilon\in(\varepsilon_0-h,\varepsilon_0+h)}{\cup}\tilde{A}_\varepsilon(t)$ possesses the relative compactness in space $\mathcal{X}$.*

**Proof.** According to the representation of the pullback attractor $\tilde{\mathcal{A}}_\varepsilon(t) = \{\tilde{A}_\varepsilon(t)\}_{t\in\mathbb{R}}$, we get

$$S_\varepsilon(t, t-\varsigma)\tilde{D}_\varepsilon(t-\varsigma) \subset \tilde{D}_\varepsilon(t), \ \forall \varsigma > T, \ t \in \mathbb{R}$$

and for any $\varepsilon \in (\varepsilon_0 - \hbar, \varepsilon_0 + \hbar)$, there always holds that $\tilde{A}_\varepsilon(t) \subset \tilde{D}_\varepsilon(t)$.

Suppose that $\{\varepsilon_n\}_{n \in N} \subset (\varepsilon_0 - \hbar, \varepsilon_0 + \hbar)$, and $u_n(t)$ is the system solution to (1) with the initial datum $u_n(t - \varsigma) \subset \tilde{D}_{\varepsilon_n}(t - \varsigma)$ $(n = 1, 2, \cdots)$, then

$$
\frac{d}{dt}(u_m - u_n) + \nu A(u_m - u_n) + B(u_m) - B(u_n) + G(u_m) - G(u_n)
$$
$$
= P(\varepsilon_m - \varepsilon_n)g. \tag{13}
$$

Letting $w = u_m - u_n$ and multiplying (13) by $Aw$, we get

$$
\frac{d}{dt}\|w\|^2 + 2\nu|Aw|^2
$$
$$
\leq \nu|Aw|^2 + \frac{3}{\nu}(|(\varepsilon_m - \varepsilon_n)g|^2 + |G(u_m) - G(u_n)|^2 + |B(u_m) - B(u_n)|^2), \tag{14}
$$

where

$$
|B(u_m) - B(u_n)|^2 = |B(w, u_m) - B(u_n, w)|^2
$$
$$
\leq \int_\Omega |w|^2 |\nabla u_m|^2 dx + \int_\Omega |u_n|^2 |\nabla w|^2 dx \leq C\|w\|_4^2 \|\nabla u_m\|_4^2 + \|u_n\|_{L^\infty}^2 \int_\Omega |\nabla w|^2 dx
$$
$$
\leq C\|\nabla w\|^2 \|Au_m\|^2 + C\|u_n\|_{L^\infty}^2 \|w\|^2 \leq C(|Au_m|^2 + |Au_n|^2)\|w\|^2, \tag{15}
$$

$$
|G(u_m) - G(u_n)|^2
$$
$$
\leq C \int_\Omega ((|u_m|^{\gamma-1} - |u_n|^{\gamma-1})^2 u_n^2 + |u_m|^{2\gamma-2}(u_m - u_n)^2) dx
$$
$$
\leq C \int_\Omega ((|u_m|^{\gamma-2} + |u_n|^{\gamma-2})^2 (u_m - u_n)^2 u_n^2 + |u_m|^{2\gamma-2}(u_m - u_n)^2) dx
$$
$$
\leq C \int_\Omega (|u_m|^{2\gamma-2} + |u_n|^{2\gamma-2})(u_m - u_n)^2 dx
$$
$$
\leq C(\|u_m\|_{L^\infty}^{2\gamma-2} + \|u_n\|_{L^\infty}^{2\gamma-2}) \int_\Omega (u_m - u_n)^2 dx
$$
$$
\leq C(|Au_m|^{\gamma-1} + |Au_n|^{\gamma-1})\|w\|^2. \tag{16}
$$

Using (15) and (16) in (14), we have

$$
\frac{d}{dt}\|w\|^2 + \nu|Aw|^2
$$
$$
\leq C(|Au_m|^2 + |Au_n|^2 + |Au_m|^{\gamma-1} + |Au_n|)\|w\|^2 + \frac{3}{\nu}(\varepsilon_m - \varepsilon_n)|g|^2, \tag{17}
$$

and

$$
\frac{d}{dt}(e^{\delta t}\|w\|^2) + (\nu - \frac{\delta}{\lambda_1})|Aw|^2
$$
$$
\leq C(|Au_m|^2 + |Au_n|^2 + |Au_m|^{\gamma-1} + |Au_n|)e^{\delta t}\|w\|^2 + \frac{3}{\nu}|\varepsilon_m - \varepsilon_n|e^{\delta t}|g|^2. \tag{18}
$$

Integrating (18) with respect to $t$ and using the estimate of $|Au|$ in Lemma 4 leads to

$$
\begin{aligned}
&\|u_m(t) - u_n(t)\|^2 \\
&\leq e^{-\delta\varsigma}(\|u_m(t-\varsigma) - u_n(t-\varsigma)\|^2 + \frac{3}{\nu}(\varepsilon_m - \varepsilon_n)e^{-\delta t}\int_{-\infty}^{t} e^{\delta s}|g|^2 ds \\
&\quad + C\int_{t-\varsigma}^{t}(|Au_m|^2 + |Au_n|^2 + |Au_m|^{\gamma-1} + |Au_n|^{\gamma-1})\|w\|^2 ds \\
&\leq Ce^{-\delta\varsigma}R_{\varepsilon_0}^2(t-\varsigma) + \frac{3}{\nu}(\varepsilon_m - \varepsilon_n)e^{-\delta t}\int_{-\infty}^{t} e^{\delta s}|g|^2 ds \\
&\quad + C_{t,\varsigma}\int_{t-\varsigma}^{t}\|u_m(s) - u_n(s)\|^2 ds.
\end{aligned}
\tag{19}
$$

According to Lemmas 3 and 4, we show in $L^2(t-\varsigma, t; \mathcal{V})$ that

$$
u_m(\cdot) \to u(\cdot),
$$

and

$$
\lim_{m\to\infty}\lim_{n\to\infty}\int_{t-\varsigma}^{t}\|u_m(s) - u_n(s)\|^2 ds = 0.
$$

Further, the sequence $\{\varepsilon_n\}_{n\in\mathbb{N}} \subset (\varepsilon_0 - \hbar, \varepsilon_0 + \hbar)$, thus there exists a Cauchy subsequence such that

$$
\lim_{m\to\infty}\lim_{n\to\infty}(\varepsilon_m - \varepsilon_n)e^{-\delta t}\int_{-\infty}^{t} e^{\delta s}|g|^2 ds = 0.
$$

We denote

$$
\Phi(t-\varsigma, \varepsilon_m, \varepsilon_n, u_m, u_n) = \frac{3}{\nu}(\varepsilon_m - \varepsilon_n)e^{-\delta t}\int_{-\infty}^{t} e^{\delta s}|g|^2 ds + C_{t,\varsigma}\int_{t-\varsigma}^{t}\|u_m(s) - u_n(s)\|^2 ds,
$$

then $\Phi(t-\varsigma, \cdot, \cdot, \cdot, \cdot)$ is a contractive function with domain

$$
(\varepsilon_0 - \hbar, \varepsilon_0 + \hbar) \times (\varepsilon_0 - \hbar, \varepsilon_0 + \hbar) \times \bigcup_{\varepsilon\in(\varepsilon_0-\hbar,\varepsilon_0+\hbar)}\tilde{D}_\varepsilon(t-\varsigma) \times \bigcup_{\varepsilon\in(\varepsilon_0-\hbar,\varepsilon_0+\hbar)}\tilde{D}_\varepsilon(t-\varsigma),
$$

and according to Lemma 3, the proof is complete. $\square$

From Lemmas 5–7, Theorem 5 is obtained.

## 5. Further Study

Our work has studied the robustness of a three-dimensional Navier–Stokes system with perturbation force term and increasing damping term via the upper semicontinuity of system attractors on a bounded smooth domain. However, the related research on a non-smooth domain is still open, and we will take time to study this topic in the future.

**Author Contributions:** Conceptualization, methodology, K.S. and J.C.; writing—original draft preparation, K.S.; software, writing—review and editing, funding acquisition, J.C. All authors have read and agreed to the published version of the manuscript.

**Funding:** This research was funded by the Natural Science Foundation of Henan Province, China (Grant number 212300410164), and Key Research Project of Institutions of Higher Education of Henan (Grant number 23B110003).

**Data Availability Statement:** Not applicable.

**Conflicts of Interest:** The authors declare no conflict of interest.

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
