# Peer review of "Robustness of 3D Navier–Stokes System with Increasing Damping"

_applsci, doi:10.3390/app13031255_

Round 1
Reviewer 1 Report
The authors deal with the 3D Navier-Stokes system with a non-autonomous perturbation term and a damping term. Under some assumptions on the model data, they established the existence and robustness of pullback attractors.
The topic of the work is very interesting and, as far as I know, the presented results are original. However, this article is purely mathematical and does not touch upon any applied aspects. But discussion of real application is highly desirable, since the article has been submitted to the journal "Applied Sciences". Moreover, main physical aspects should be discussed, in particular, one can describe the physical meaning of the damping term as it is done in the paper
Baranovskii E. S., Artemov M. A. Model for aqueous polymer solutions with damping term: Solvability and vanishing relaxation limit, Polymers, 2022, 14 (18), Article ID 3789, https://doi.org/10.3390/polym14183789
Finally, I recommend adding the conclusion section to the manuscript and eliminate formulas in the abstract section.
Author Response
Dear Professor,
Many thanks for your time and effort to our article, your advice and the recommended paper are of great help and inspiration to us. In the introduction, we add a lot of background introductions and physical aspects of the increasing damping term, especially in the fluid system within porous media. And we eliminate formulas in the abstract section and add some more sentences about the result.
Kind regards,
Keqin Su

Reviewer 2 Report
See attached report.

Author Response
Dear Professor,
Many thanks for your time and effort to our article, your advice is of great help and inspiration to us. We fully agree with the comments made and make modifications accordingly. Also, we add a lot of background introductions and physical aspects of the increasing damping term, especially in the fluid system within porous media, and add some more sentences about the result in the abstract section. We check the presentation and language carefully and provide the future direction finally.
Kind regards,
Keqin Su

Reviewer 3 Report
See the report attached

Author Response

(The authors gave the same response as above.)

Reviewer 4 Report
The authors have studied the robustness of the 3D Navier-Stokes system with increasing damping.
The paper can be improved by addressing the following comments.
1) The abstract should be improved by adding some more sentence about results.
2) Introduction should be started form a hot research topic such partial differential equations and its application. The author can describe the following works
https://doi.org/10.1016/j.joes.2022.04.010
https://doi.org/10.1016/j.physleta.2022.128393
3) All citations should be in proper sequence.
4) The presentation and language should be checked and improved.
5) The conclusion and future direction should be added.
Author Response
Dear Professor,
Many thanks for your time and effort to our article, your advice and the two recommended papers are of great help and inspiration to us. In the introduction, we add a lot of background introductions and physical aspects of the increasing damping term, especially in the fluid system within porous media. We eliminate formulas in the abstract section and add some more sentences about the result. We check the presentation and language carefully and provide the future direction in the last part.
Kind regards,
Keqin Su

Round 2
Reviewer 1 Report
I am satisfied with the answer and revisions that the authors have performed in their manuscript. The results of the work will be interesting and useful to specialists in the subject area. I recommend this paper for publication in Applied Sciences.
My minor suggestions are as follows:
1) Line 136: From Lemma 5, Lemma 6 and Lemma 7 --> From Lemmas 5-7
2) Line 154: Studies in ... --> Book Series: Studies in ...
3) Line 157: 1–17 --> 3789
Reviewer 4 Report
The author have addressed my comments in a professional way. I recommend the paper for publication in its current form.
